# Comparative Study on Life-Cycle Assessment and Carbon Footprint of Hybrid, Concrete and Timber Apartment Buildings in Finland

**DOI:** 10.3390/ijerph19020774

**Published:** 2022-01-11

**Authors:** Roni Rinne, Hüseyin Emre Ilgın, Markku Karjalainen

**Affiliations:** Faculty of Built Environment, School of Architecture, Tampere University, P.O. Box 600, FI-33014 Tampere, Finland; roniorinne@gmail.com (R.R.); markku.karjalainen@tuni.fi (M.K.)

**Keywords:** comparative study, life-cycle assessment (LCA), carbon footprint, hybrid, concrete, timber, apartment building, Finland

## Abstract

To date, in the literature, there has been no study on the comparison of hybrid (timber and concrete) buildings with counterparts made of timber and concrete as the most common construction materials, in terms of the life cycle assessment (LCA) and the carbon footprint. This paper examines the environmental impacts of a five-story hybrid apartment building compared to timber and reinforced concrete counterparts in whole-building life-cycle assessment using the software tool, One Click LCA, for the estimation of environmental impacts from building materials of assemblies, construction, and building end-of-life treatment of 50 years in Finland. Following EN 15978, stages of product and construction (A1–A5), use (B1–B6), end-of-life (C1–C4), and beyond the building life cycle (D) were assessed. The main findings highlighted are as following: (1) for A1–A3, the timber apartment had the smallest carbon footprint (28% less than the hybrid apartment); (2) in A4, the timber apartment had a much smaller carbon footprint (55% less than the hybrid apartment), and the hybrid apartment had a smaller carbon footprint (19%) than the concrete apartment; (3) for B1–B5, the carbon footprint of the timber apartment was larger (>20%); (4) in C1–C4, the carbon footprint of the concrete apartment had the lowest emissions (35,061 kg CO_2_-e), and the timber apartment had the highest (44,627 kg CO_2_-e), but in D, timber became the most advantageous material; (5) the share of life-cycle emissions from building services was very significant. Considering the environmental performance of hybrid construction as well as its other advantages over timber, wood-based hybrid solutions can lead to more rational use of wood, encouraging the development of more efficient buildings. In the long run, this will result in a higher proportion of wood in buildings, which will be beneficial for living conditions, the environment, and the society in general.

## 1. Introduction

The construction industry is one of the leading producers of greenhouse gas (GHG) emissions [1], contributing to more than 35% of global energy use and about 40% of energy-related CO_2_ emissions [2,3,4,5]. Additionally, in the European Union, building construction consumes 40% of materials and 40% of primary energy and produces 40% of annual waste [6]. Therefore, the construction industry represents a significant potential for reducing CO_2_ emissions and tackling climate change [7,8].

There are currently two main approaches adopted in the construction industry to reduce environmental impacts: (1) appropriate material selection; and (2) the optimization of energy use throughout the building’s service life [9,10,11,12]. GHG emissions from buildings can be significantly affected by the choice of building materials. In this regard, according to the calculation made by Hafner [13] on the life-cycle assessment (LCA), depending on the building construction standard, the operation stage accounts for 45% to 80% of the total CO_2_ emissions, in which the materials used account for 20% to 55% of total CO_2_ emissions. On the other hand, it was reported that the application of alternative additives/materials or techniques/systems can reduce CO_2_ by up to 90% [14].

Environmental effects should also be taken into account when applying conventional building materials such as steel and concrete, which are used in large quantities. While the steel industry accounts for about 9% of the direct emissions from global fossil fuel use [15], concrete production accounts for about 8% of the world’s CO_2_ emissions [16,17]. Recent research has focused on the development of materials with low CO_2_ emissions that can mitigate climate change by reducing these emissions or storing carbon in the long term [18,19]. In this sense, wood buildings are characterized by the concept of lower-carbon construction than non-wood buildings [20,21,22,23,24], and wooden construction represents a lower embodied energy consumption compared to steel and concrete production [25]. Wooden structures provide significant advantages of tackling climate change, because wood not only can be used as an alternative to other materials to reduce GHG emissions, but also has distinctive features such as storing large amounts of carbon in the structure [26,27]. Besides being used as a building material during the construction of a building, wood can be reused as a raw material for other structures after the building’s service life or, as a last resort, burned instead of fossil fuels [28,29,30].

Engineered wood products (EWPs) such as cross-laminated timber (CLT) are used in increasingly demanding applications to meet sustainable construction challenges [31,32]. Besides the many advantages of CLT, such as high strength-to-weight ratios, low carbon, and high thermal insulation, it also allows residential, commercial, and institutional multistory building construction in a cost-effective way [33,34,35]. Additionally, CLT is mainly structural but can also be used as a facade material and a secondary building material, e.g., floor and non-structural walls [19]. In the literature, many studies focus on the technological aspects of EWPs, their use in construction, and different building solutions [36,37,38,39,40,41,42,43,44,45,46,47,48,49,50]. Several studies address wood as a structural material in buildings from the perspectives of key professionals (e.g., [51,52,53,54,55,56,57,58,59]) and consumers or residents (e.g., [60,61,62,63,64,65]).

In the last decade, the LCA for EWPs has become an important research focus [7,66,67]. Among them, the results of Robertson’s cradle-to-grave LCA analysis [68] showed that maximizing the use of wood in buildings is over 70% more advantageous than using concrete in mitigating the impact of global warming. Darby [69] reported that CLT has remarkable effects on the reduction of GHG emissions, regardless of whether the building is disposed of after reaching the end of its service life. Skullestad [70] found that replacing steel and concrete with wood (CLT and glulam) results in GHG reductions ranging from 34% to 84% for all four types of building construction. Similarly, the results of the study by Milaj [71] on six examples of commercial buildings using cradle-to-grave LCA showed that the use of wood, instead of steel and concrete, results in an average of a 60% reduction in GHG emissions. Gu and Bergman [72] conducted an entire building LCA in a four-story educational building using large amounts of CLT roofs, floors, and staircase wall panels to develop the first Environmental Building Declaration (EBD) in the USA, and the building eventually earned LEED (Leadership in Energy and Environmental Design) credits to mitigate the LCA impact. Chen et al. [7] compared the results of a cradle-to-grave LCA for a 12-story building made of CLT and a functionally equivalent reinforced-concrete (RC) building following the EN 15978 framework. For the material resource efficiency, they found that the total mass of the CLT building was 33.2% less than that of the alternative RC building, there was a 20.6% reduction in concrete carbon achieved for the CLT building compared to for the RC building, and the emissions from the CLT building were 70% lower than the RC building. Oladazim et al. [18] conducted a case study to analyze the life cycle environmental impacts of two multi-story residential buildings, one with an RC and the other with a steel frame, in Iran. The results showed the total amount of pollution by the RC building at various stages was 38% higher than by the steel building, and the steel frame selection in RC buildings was more environmentally friendly than the building industry concrete frame. Liang et al. [73] made a comparative LCA of a 12-story mixed–use building constructed predominantly from solid timber (CLT and glulam) and a similar concrete building. The results indicated that the solid timber building had reductions of 18%, 1%, and 47% in the global warming, ozone depletion, and eutrophication impact categories, respectively, and the use of solid timber significantly reduced the carbon footprint of the building. Ryberg et al. [74] performed an LCA on four buildings in Greenland, namely an RC building, a CLT building, a timber-frame building, and a renovation of an existing concrete building, to assess environmental impacts in the midpoint indicator and the overall damage to human health, the ecosystem quality, and resources. The results highlighted that the refurbishment of existing buildings had the lowest environmental impact on all impact categories, and the difference in environmental impacts between new building types was generally small, while the CLT and wood-frame buildings still had the best environmental performance. Liang et al. [75] examined a high-rise mass-timber building in the Pacific Northwest using LCA and life-cycle cost analysis approaches to assess the life-cycle environmental and economic performances and compared these results to for a concrete building of the same design. Cradle-to-grave (modules A–C) LCA results, which have been in operation for over 60 years, showed that the mass-timber building outperformed the concrete building in terms of global warming (1.6% lower) and other environmental impacts.

In Finland, the use of timber in buildings is most common in single-family homes (80% of buildings made with wooden frames) and row houses (60% of buildings made with wooden frames) [76]. Despite this strong timber building culture and the vast forest resources in Finland, the use of timber in multi-story buildings such as apartments is still under development, and its market shares are still low [77,78,79]. However, the acceleration of wooden multi-story buildings as a set of innovative building technologies has gained political support and public attention in Finland as well as in other forest-rich European countries [80]. Since the 1990s, strong emphasis has been placed on increasing the development of the wood construction business, removing regulatory barriers and increasing the business development of wood construction companies [79,81]. On the other hand, the Finnish concrete industry, which has led the building construction market for many years, due to its routine, strong presence, and networking in both public and private organizations [82,83], has faced new challenges such as eco-friendly standards, people’s perception of sustainability, and the coziness of the building as in wooden multi-story apartment buildings [84,85].

As noted above, EWPs, e.g., CLT, could have excellent opportunities to emerge as a potential competitor for concrete multi-story buildings, but the Finnish construction industry also puts great trust in hybrid structures, which includes the selection of the best components and materials to achieve sustainable construction principles [86]. From this point of view, the encounter of wood with other materials, especially concrete, may not be seen as a positive phenomenon [87]. After all, competition does not benefit the continued development of new construction technologies and materials, and cooperation between industries will bear more fruit [88]. In this sense, hybrid structures should be encouraged, thus maximizing the advantages of different materials [89]. In addition, concerning the changing trends in timber construction affecting the entire business, as part of the rise of the green building concept, the contemporary tendency in timber construction increasingly includes the use of hybrid structures (e.g., wood and concrete or steel combinations) [25].

It is worth noting here that there is a growing interest in measuring and reducing environmental burdens with climate change and other adverse environmental impacts. At this point, the problem of how to measure and reduce environmental loads comes to the fore [90]. Recently, academics, organizations, and others have sought to develop concepts and procedures that measure environmental sustainability, where the environmental footprint, which is one of the important topics discussed at the Habitat Conferences [91,92], is becoming increasingly popular and playing an important role in sustainability assessment and research [93,94]. Environmental footprints are quantitative measures of human use of natural resources [95]. Footprints are divided into environmental, economic, and social footprints and combined environmental, social, and/or economic footprints [96]. The concept of footprint is derived from the ecological footprint idea introduced by Rees [97] and Fang et al. [98]. In recent years, the carbon footprint has been used almost exclusively as an environmental protection indicator (e.g., [99,100,101,102]).

As seen in the literature review above, there are many LCA- and carbon footprint-based studies comparing timber to traditional building materials such as concrete. However, no study has been found in the literature on the comparison of timber buildings with both concrete as the most common construction material and a hybrid construction material (timber and concrete) in terms of LCA and carbon footprint. With this study, it was aimed to fill this gap in the literature.

Overall, this paper examined the environmental impacts of a five-story hybrid apartment building compared to those of timber and a more traditional reinforced concrete counterpart in whole-building LCA using the software tool, One Click LCA, for the estimation of environmental impacts (i.e., carbon footprint) from building materials of assemblies, construction, and building end-of-life treatment of 50 years in Finland. In this paper, the (wood-based) “hybrid building” refers to the building with predominantly reinforced concrete load-bearing structures, except for the top floor (timber-framed), and with a timber-framed and -clad exterior facade. The results are believed to provide critical stakeholders with a roadmap in their pursuit of a better material selection for multi-story construction to minimize environmental burdens and mitigate climate change while considering the potentials of hybrid construction.

In this research, wood or timber refers to EWPs [103,104] such as CLT (a prefabricated multi-layer EWP, manufactured from at least three layers of boards by gluing their surfaces together with an adhesive under pressure), laminated veneer lumber (LVL; made by bonding together thin vertical softwood veneers with their grains parallel to the longitudinal axis of the section, under heat and pressure), and glue-laminated timber (glulam) (abbreviated as GL; made by gluing together several graded timber laminations with their grains parallel to the longitudinal axis of the section).

The remainder of this paper was structured as follows: First, the explanations of the materials and methods used in the study were provided. This was followed by the results and discussion. Finally, the conclusions of the study were presented with suggestions for future research.

## 2. Materials and Methods

### 2.1. Hybrid Building Design

Although wood is an excellent building material in many ways and it is currently the best material in building construction toward diminishing CO_2eq_ emission compared to the alternative building materials such as concrete or brick, wood is a nature-created material and, like other building materials, requires engineering to adapt to construction conditions. It, therefore, makes sense that there is room to improve some of the weaker properties of wood as a building material, thereby making it more competitive against other less climate-friendly materials. Craftsmen, engineers, and architects from past to present have always combined different materials, often to take advantage of their best properties. This approach enables the creation of an end product with better properties than the materials of which it is composed. In this sense, smart timber-based hybrid solutions can lead to the more rational use of wood, fostering the development of more efficient buildings and reinforcing wood’s weak properties, e.g., sagging, vibration, and acoustic considerations.

Considering the facts mentioned above, in this paper, a hybrid building was designed as a case study on a plot of the land at Laurinmäenkuja 3 in Lassila, Helsinki, Finland (Figure 1). On this plot was a four-story brick-clad office building that was to be demolished from a new construction road (Figure 2).

A necessary site planning was made according to a legally binding zoning plan. Three apartments that would replace the office building in the city plan and an indoor parking lot were zoned under the entire courtyard deck. Some critical city planning regulations for the site were as following: (i) the building right of the plot is 7250 m^2^; (ii) the residential building in Laurinmäenkuja must have a maximum of 5-story, and two more remote houses can have a maximum of 8 stories; (iii) on the Laurinmäenkuja side of the building (roadside), 2/3 of the apartments and all apartments in other buildings must have a balcony or terrace; (iv) there must be an external staircase between the yard deck and the street; and (v) the site should produce renewable energy, e.g., solar energy.

In the architectural design, the starting point of the plan was to make a P1 fire class apartment using as much wood as possible within the framework of fire regulations, which meant that the load-bearing structures must be concrete, except for the top floor. It is worth mentioning here that there is an exception in the Finnish fire code that allows the top floor of a building of class P1 to have timber load-bearing structures if the building has no more than 8 stories. Additionally, an apartment building in Finland can be built in three fire classes: P0, P1, and P2 [105]. The categories P1 and P2 are based on the reference values in the fire code. P1 load-bearing structures in the fire class must be a non–combustible material, mostly concrete. For this reason, wooden flats fall into the P2 category. The P0 category is for buildings calculated based on the default fire development. Fire class P0 is used when it is desired to deviate from the table values.

The five-story hybrid building, which was designed in detail and stretched along the street, is shown in Figure 3. All interiors of the apartments were designed following the regulations and standards issued for housing design in Finland [106]. The building that ran along the street was chosen for a closer look, as it was more diverse in its starting points (Figure 4). It had a ground floor that opened to both the street and the car park. Balconies that opened to the street side should also be cantilevered balconies, which were structurally more difficult than implementing a floor-to-ceiling balcony zone. In addition, plenty of wood was used on the interior surfaces, and the P1 fire class limits the use of wood on these surfaces, especially for escape routes. In addition, rooftop solar panels were employed to comply with the renewable energy requirements of the city planning regulation, as seen in Figure 3b.

The building was also designed in more detail in terms of structural engineering. In the structural design, the main load-bearing system (e.g., shear walls, columns, and floor slabs) of the building was designed as reinforced concrete, except for the top floor, as seen in Figure 5. CLT was used as an exterior cladding to maximize the use of wood on the facade, and P1 class in accordance with the fire regulations was provided because it was desired to be left exposed due to aesthetic concerns. On the other hand, the protruding balconies were structurally supported by a load-bearing concrete inner shell (Figure 6). Additionally, outer walls had delta beams connected to concrete shear walls or columns. The hollow-core slabs were supported at one end by a load-bearing concrete wall, and at the other end, a supporting surface acted as a delta beam flange (Figure 7). Double-sided delta beams were used for cantilevered balconies (Figure 8), flanged and perforated on both sides. The supporting steel brackets for the cantilever balcony were threaded through holes in the delta beam and attached to the building’s frame. It is worth noting here that as already mentioned, the facades were completely fire-retardant. In any case, the facades of the balconies had fire protection under the fire code, which complicated the construction of the facade and reduced the order quantity of a single material size. The use of non-combustible materials also necessitated the use of fire curtains and eaves in facade structures.

### 2.2. Goal and Scope

This study aimed to examine the environmental impacts of a 5-story hybrid apartment building compared to those of timber and reinforced concrete counterparts with the whole-building LCA using the software tool ‘One Click LCA’ (Helsinki, Finland), for the estimation of environmental impacts (carbon footprint) from building materials of assemblies, construction, and building end-of-life treatment of 50 years in Finland.

Only a more specially designed five-story apartment building (Figure 3a) was considered in this comparison. The courtyard decks from other buildings were not included. The impact of the demolition of an existing building or earthworks on the plot on emissions as well as the effect of fixed furniture, furniture, or other equipment was also not taken into account.

The comparison was made by taking a replica of the hybrid apartment building and switching to different building types. It was assumed that the U values of the building types did not change. A summary of the features of the reference apartment and the changes made were listed below:

Concrete apartment

The outer walls were load-bearing concrete sandwich elements. The intermediate floor structure was the same as in that of the hybrid apartment.Concrete tiles were used instead of CLT volume elements on the balconies.The street facade had a brick tile surface, making zoning requirements more flexible as in the hybrid apartment. The courtyard facade was made of lightweight concrete.

Timber apartment

The framework was designed based on the CLT element technology. LVL-ribbed tiles were used on the mid floors.It was the entire ground layer of the concrete structure to prevent the building from settlement differently.A sprinkler system was added.Balconies were the same CLT volume elements as in the hybrid apartment.

As seen in Figure 9, the system boundary was defined as a cradle-to-grave boundary, which included the product stages (A1–A3), the construction stages (A4–A5), the use stages (B1–B6), and the end–of–life stages (C1–C4) [106]. System expansion was used to account for the net benefits associated with energy recovery from materials such as wood burning, as well as the net benefits associated with reuse, recycling, and recovery potential (D).

For this study, B7—operational water use—was excluded from the comparative LCA. Although the global warming impact from the whole-building LCA comes mainly from operational energy, such as electricity and natural gas used during the building’s lifetime, three comparison buildings were designed as functionally equivalent. This analysis mainly focused on the impacts resulting from materials and other activities.

It is also worth noting here that the results for A1–A5, B1–B5, and C1–C4 were produced by combining the quantity and material data from Autodesk Revit with One Click LCA’s Environmental Product Declarations database. On the other hand, the results for B6 were generated by combining results from the E-value calculator [107] for different energy forms in Finland provided by the Finnish Ministry of Environment with the emission coefficients [108].

### 2.3. Software Tool

According to the EN 15t978 standard [109], carbon footprint calculation was made using One Click LCA software, a web and cloud-based computing tool. This is automated LCA software that helps calculate and reduce the environmental impact of building and infrastructure projects, products, and portfolios [110]. One Click LCA provides a plug-in for Autodesk Revit that allows importing quantity and material information directly from the Revit model. The buildings examined in this study were designed and modeled in Autodesk Revit. The program uploads the material information to the cloud service and retrieves the materials corresponding to the environmental notifications from its database. Most materials information is stored in software that can automatically retrieve an appropriate environmental statement, but for some materials, the user needs to search the database for an appropriate environmental statement.

One Click LCA is currently one of the most advanced tools available on the market for a simplified LCA approach [111]. It allows the use of international green building certificates and databases as well as building information modeling (BIM)-based workflows for all evaluated certificates. One Click LCA complies with EN/ISO standards and more than 40 certifications such as EN 15978, which covers impacts from production, transportation, construction, use, and demolition as well as operational energy and water use, EN 15804, EN 15942, ISO 21931–1, ISO 21929–1, ISO 21930, BREEAM, LEED, HQE, and C+E–. Moreover, it integrates with building information models in the IFC2x3, IFC4, Revit, ArchiCAD, and Tekla Structures formats (see Table 1). Additionally, it integrates with all energy modeling software supporting the gbXML format (including DesignBuilder and IES-VE), Microsoft Excel, and other data formats. There is also a direct integration into the IES-VE software [110]. In this study and many other studies (e.g., [112,113,114,115,116,117]), One Click LCA was used because of the abovementioned features, ease of use, and fast and accurate results.

## 3. Results 

### 3.1. Initial Values and Assumptions

Initially, the energy consumption of each building was calculated using the E-value calculator [107]. On the counter, the building class was defined as an at least three-story apartment building. Using the net heated area of the building, the number of floors, the data of the structures, and the architectural model taken from Autodesk Revit, the outer walls, upper and lower floors, and exterior door areas were provided. In addition, thermotechnical values were provided, and interface counter defaults were used to evaluate the cold bridge effect.

Mostly default counter values were used for ventilation. Mechanical “normal efficiency ventilation” was chosen as the ventilation type. It was determined that the ventilation post-heating battery was connected to the heating system. The q50 value of the air leakage number was determined as 1.5. District heating and heat distribution systems were chosen as the heating system with water circulation underfloor heating. The building was assumed to be a 300 L hot water storage tank, and hot water circulation and transmission lines rotating in a protective tube were chosen as the type.

Self-sufficient electricity generation from solar panels on the building’s roof was provided. The roof of the building had 84 solar panels with a size of 1.6 m^2^ facing south and at an angle of 30 degrees with respect to the horizontal. The rated power of a single solar panel was 300 Wp. The south is the best weather direction for solar panels, and 99% of the electricity was produced by the solar panel at the optimum angle of 30 degrees. With a peak power of 3 kW, the solar panel system produced around 2500 kWh of electricity per year in Finland. With this information, the peak power of the photovoltaic system of the building was calculated as 25,200 Wp, and the annual electricity production was calculated as 21,000 kWh. This had a significant impact on reducing the use of purchased electricity.

All buildings to be compared were identical for their initial values. Only the solidity of the building envelope was changed. The results on the energy use were reported in Table 2. These numbers were entered into One Click LCA software. In comparison, electricity consumptions were the same in all apartments.

On the One Click LCA side, the lifetime of a building was defined as 50 years. The Scandinavian average was chosen for transport distance, which affected the module A4 result. Material emissions were adjusted to meet Finnish production conditions. The production emissions from different materials may vary by country, for example, based on the carbon footprint of electricity available in the country, where emission values were adapted to Finnish conditions and the Finnish electricity generation carbon footprint. The densities of concrete and reinforced concrete were taken as 2400 kg/m^3^ and 2500 kg/m^3^, respectively. The carbon footprint calculation for the construction work A5 was also based on square meters relative to the Scandinavian average. The influence of building materials was not reflected in the calculation result of module A5.

### 3.2. Results of Comparison

The results of carbon footprint calculation are presented in Table 3 and Figure 10. Energy use contributed most to the carbon footprint of every building followed by product stages A1–A3. The timber apartment building had the lowest life-cycle emissions. The hybrid and concrete apartment buildings were similar, but the concrete apartment building had a slightly smaller carbon footprint. However, the building life-cycle value (module D) of the hybrid apartment was greater but not included in carbon footprint.

## 4. Discussion

As highlighted in Table 3 and Figure 10, at the product stage, the timber building had the smallest carbon footprint (28% less than the hybrid building). The hybrid and concrete apartments were close together, but the carbon footprint of the hybrid apartment was slightly smaller in modules A1–A3. Replacing concrete structures under construction with timber helped reduce the carbon footprint of the hybrid apartment. However, the composite frame required extra concrete pillars, and especially delta beams increased the carbon footprint of the product stage, which showed a very small difference from that of the concrete apartment. The photovoltaic system also increased the carbon footprint of this module for all reference buildings, but this system then appeared as a reduction in carbon footprint at B6, i.e., operational energy use.

For A4 transport, the timber apartment’s carbon footprint was by far the smallest (55% less than that of the hybrid case). The carbon footprint of the hybrid apartment was also smaller than that of the concrete building, due to the lighter materials. However, the transport stage had a smaller impact on the life-cycle carbon footprint. On the other hand, as the carbon footprint of construction work in module A5 was based on the floor area value, this calculation was not different in this module.

In B1–B5, i.e., the use of products and the refurbishment module, the hybrid and concrete apartments had very close values. The carbon footprint of the timber apartment was larger (>20%). The most important factor in the increase was the increased use of gypsum boards, which had to be replaced from time to time.

As seen in Table 3 and Figure 10, in the results for B6—operational energy use, due to the reduced heat absorption capacity of the building, lighter building materials and the consequent increase in district heating consumption increased the carbon footprint of the hybrid and timber apartments. However, the differences were so small that the impact of the life cycle on the overall carbon footprint remained small.

The biggest surprise in the calculation results for the factor was carbon footprint of the end-of-life stages C1–C4, where the concrete apartment had by far the lowest emissions and timber apartment had the highest emissions. In this module, the concrete apartment building yielded less than half the value of that of the timber apartment building. Similarly, as shown in Figure 11, in end-of-life stages C1–C4, the carbon footprint of the timber apartment was much higher than that of the concrete and hybrid apartments.

In module D, beyond the building life cycle, the values of timber completely reversed, and it became the most advantageous building material compared to in modules C1–C4. Timber had a life cycle, and concrete had a greater potential to reduce emissions, resulting in the hybrid apartment performing much better than the concrete apartment (Figure 12). However, module D was not included in carbon footprint, unlike modules C1–C4. Thus, the carbon footprint of the concrete apartment building was slightly lower than that of the hybrid one.

More details about the different distributions of carbon footprint for resource types and materials in all three buildings are shown in Figure 13, Figure 14 and Figure 15. The share of life-cycle emissions from building services was very significant in all building types, about one-third. In the hybrid apartment case, it was interesting to observe how much delta beams affected the carbon footprint of the building. Structural steel accounted for more than 9% of the carbon footprints of all cases. In product stages A1–A3, most of the carbon footprint came from horizontal structural members, including delta beams. After all, the share of timber in the carbon footprint of the hybrid apartment was quite small. The distribution of the concrete apartment was not particularly surprising. Most of the materials’ carbon footprints came from concrete. In the timber apartment, gypsum boards accounted for the bulk of material emissions. However, the volume of gypsum boards in a building was only a small fraction compared to that of timber or concrete.

### Reducing Carbon Footprint

The delta beams of the composite frame had a major impact the on carbon footprint of the hybrid apartment building, where approximately 9% of the materials’ carbon footprint originated from these beams. Thus, the carbon footprint of the hybrid case can be reduced by reducing the number of delta beams. Two variations of the building that would make this possible are as follows.

The carbon footprints of these variations were not calculated, so the effect was not fully known. The main purpose of delta beams was to act as a load-bearing structure when the load-bearing capacity was required from the outer wall line. Therefore, the need for load-bearing external wall lines should be reduced. The first way to reduce load-bearing wall lines was to abandon cantilever balconies. Streetside cantilevered balconies can be replaced with retracted balconies. Both balconies of the courtyard facade were built by being moved from the ground, so that the outer walls did not have to carry them. Some of the outer walls can also be replaced with a hybrid structure. a load-bearing concrete inner shell, and a wooden outer shell, as shown in Figure 16. Here, version A reduced the number of delta beams to 31 running meters per floor. Only 10 load-bearing concrete walls per floor had to be added. Such a solution can be conveniently used in two blocks in a taller building, as they both have balconies carried on the ground. The number of delta beams can be further reduced by increasing the number of hybrid exterior walls. In version B, these walls were added as follows, as delta beams were not needed at all. However, in this version, concrete walls might adversely affect the aesthetic benefits of visible wood. Replacing the delta beam with a full-length concrete inner shell probably would not lower carbon footprint, either.

## 5. Conclusions

In this paper, a five-story hybrid building was designed in detail as a case study in Finland to make a comparison with timber and reinforced concrete counterparts in a whole-building LCA using One Click LCA for environmental impact estimation.

The results showed that for modules A1–A3, the timber apartment had the smallest carbon footprint (28% less than the hybrid building); in module A4, the timber building had a much smaller carbon footprint (55% less than the hybrid case), and the hybrid building had a smaller carbon footprint (19%) than the concrete apartment; for modules B1–B5, the carbon footprint of the timber building was larger than those of the others (>20%); for the carbon footprint in C1–C4, the concrete apartment had the lowest emissions (35,061 kg CO_2_-e) and timber apartment the highest (44,627 kg CO_2_-e), but in module D, timber became the most advantageous building material; the share of life-cycle emissions from building services was substantial across all building types; and in the timber apartment, gypsum boards accounted for the bulk of the material emissions.

Based on the results obtained in this paper, although the timber apartment had the lowest carbon footprint in many modules of life-cycle analysis, the hybrid (timber and concrete) apartment building also showed a notable environmental performance, especially when compared to the reinforced concrete apartment building. These results can be considered as an innovative contribution to the literature, since it is not a hybrid structure-based comparative study in terms of the life cycle assessment and the carbon footprint. Considering the environmental performance of hybrid construction as well as its other advantages over pure timber construction such as sagging, vibration, and acoustic issues, smart timber-based hybrid solutions can lead to more rational use of wood, encouraging the development of more efficient buildings. In the long run, this will give rise to a higher proportion of wood in buildings, which will be beneficial for living conditions, the environment, and the society in general. In this sense, this study, which reveals the eco-friendly potential of timber-based hybrid solutions, provides insight into and incentives to construction practitioners such as architects, developers, and contractors to integrate more timber-based hybrid solutions into their spectacular projects such as tall buildings (e.g., the 18-story and 58 m high Brock Commons Tallwood House (Vancouver, 2017) and the 24-story and 84 m high HoHo (Vienna, 2020)). In general, it is recommended to implement holistic assessments such as LCA as part of the decision process to support more environmentally friendly decisions regarding the construction industry.

Several limitations of this study should be mentioned. One of the limitations of this study was the exclusion of the impact of the demolition of an existing building or excavation on the plot on emissions, as well as the impact of fixed furniture, movable furniture, and other equipment such as windows and doors. The courtyard decks from other buildings were also excluded. Additionally, while the combination of timber and concrete was generally studied as a hybrid structure in this study, a combination of timber and steel or timber and a combination of concrete and steel can be investigated to enrich the hybridization approach.

In future research, economic and social factors affecting the choice of building materials can be also examined, where methods such as life-cycle costing [118] and social LCA [119] might be utilized. Future research may also focus on the effects of different design decisions on many aspects, e.g., variations in the column spacing, beam depth, change the amount of material, and thus life-cycle environmental impact. In this sense, modifications in design configurations that can use thinner massive wood panels, smaller foundations, and less concrete for flooring should further reduce the overall environmental impact for hybrid construction. Future work may also include other hybrid alternatives such as wood and steel. In addition, similar studies can be conducted in other Scandinavian countries to enrich the subject with comparative analysis. Finally, future research should also emphasize combining use–phase scenarios to further examine the impact of the exterior facade design on both embodied and operational energies and the interaction between the two.

## Figures and Tables

**Figure 1 ijerph-19-00774-f001:**
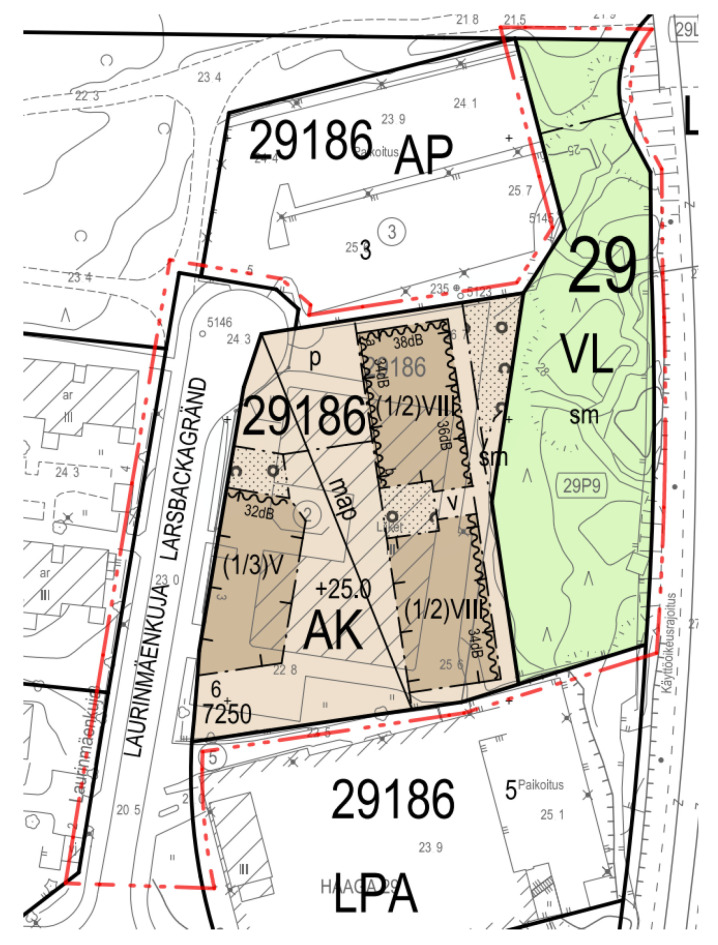
The selected plot of the land at Laurinmäenkuja 3 in Lassila, Helsinki.

**Figure 2 ijerph-19-00774-f002:**
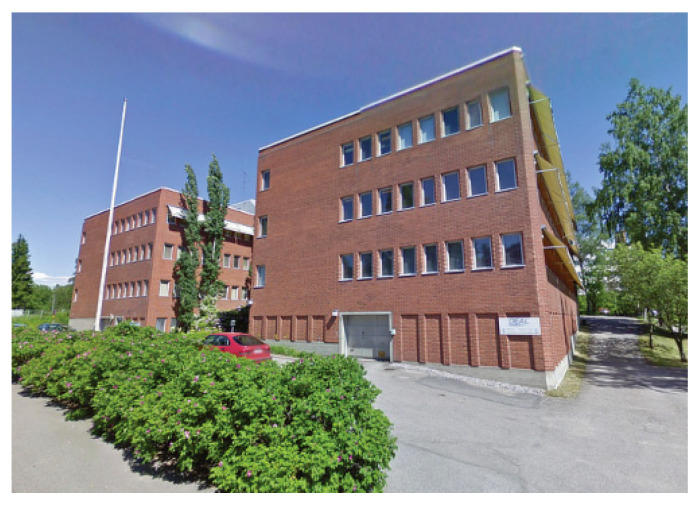
Four-story brick-clad office building to be demolished.

**Figure 3 ijerph-19-00774-f003:**
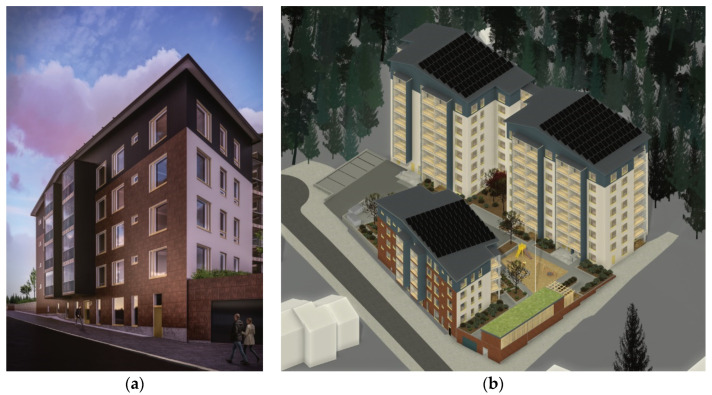
The designed five-story hybrid building in three dimensions (3D): (**a**) standalone view; (**b**) the site plan.

**Figure 4 ijerph-19-00774-f004:**
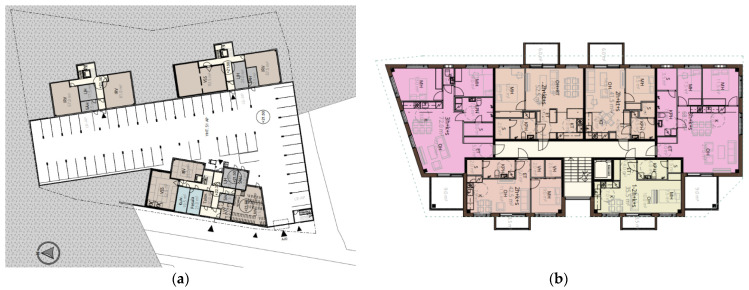
Architectural floor plans: (**a**) general site plan; (**b**) typical floor plan of the hybrid building.

**Figure 5 ijerph-19-00774-f005:**
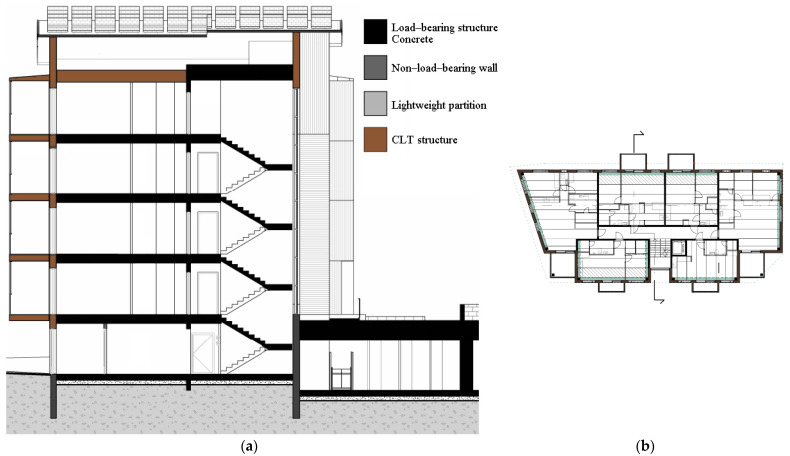
Structural section of the hybrid apartment building: (**a**) section; (**b**) key floor plan for the section.

**Figure 6 ijerph-19-00774-f006:**
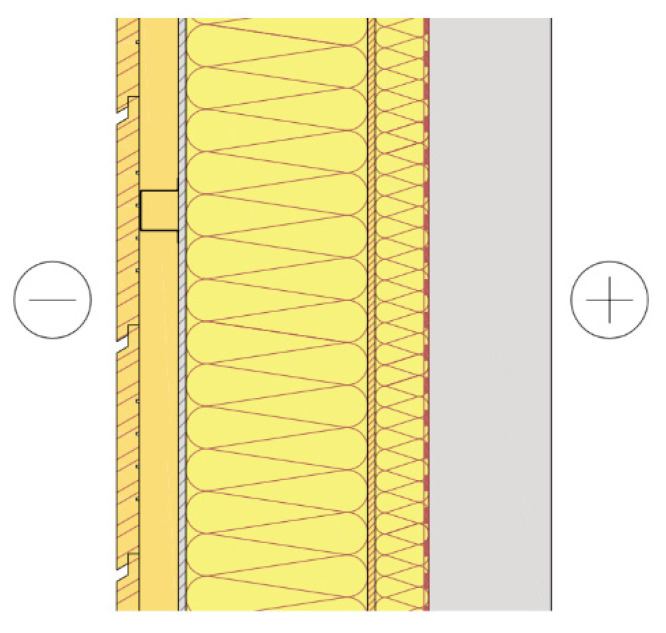
An example of a (wood (outer)–concrete (inner)) hybrid wall.

**Figure 7 ijerph-19-00774-f007:**
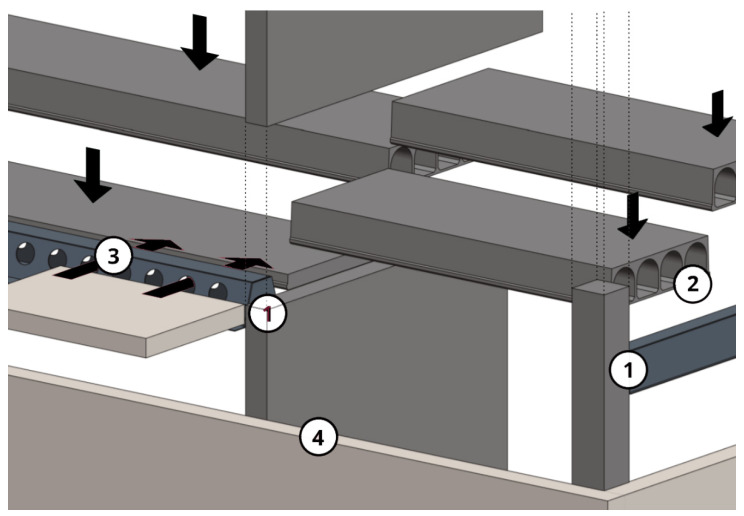
The structural principle of the plan: (**1**) delta beams were attached to the side of a concrete wall or beam with concealed brackets and with mounting plates; (**2**) hollow-core slabs rested on the flanges of the delta beam. Delta beams and hollow core slabs were joined by solder casting; (**3**) a cantilever balcony was attached to the building’s frame through the delta beam; (**4**) the cladding of the building was wood and non-load-bearing.

**Figure 8 ijerph-19-00774-f008:**
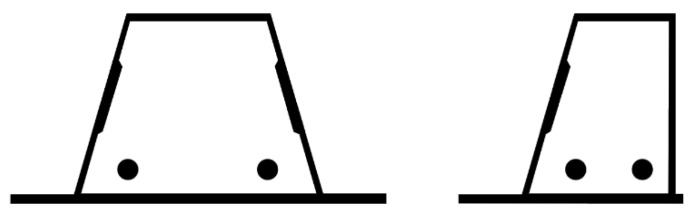
Double- and single-sided delta beam cross-sections.

**Figure 9 ijerph-19-00774-f009:**
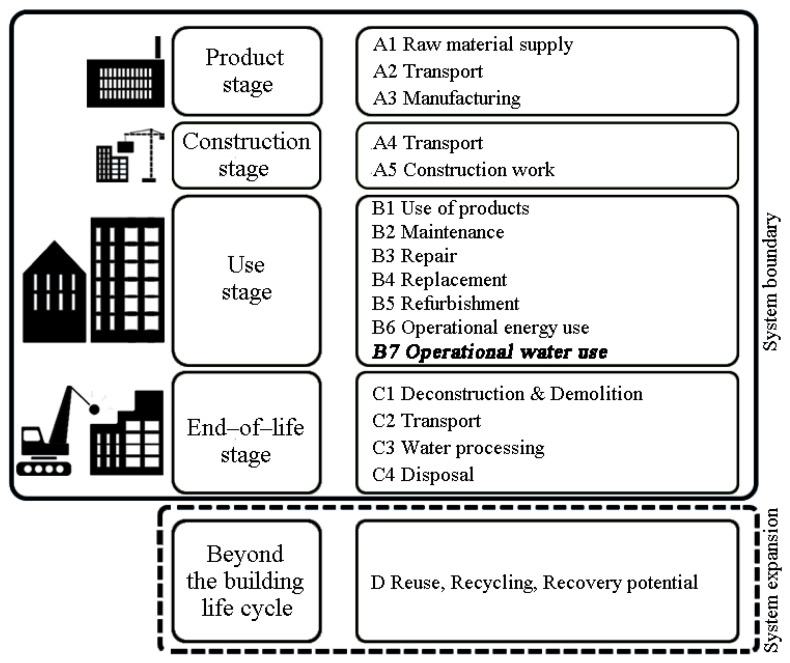
Assessment system boundary based on EN 15978. (The bold and italics stage was not included in this analysis).

**Figure 10 ijerph-19-00774-f010:**
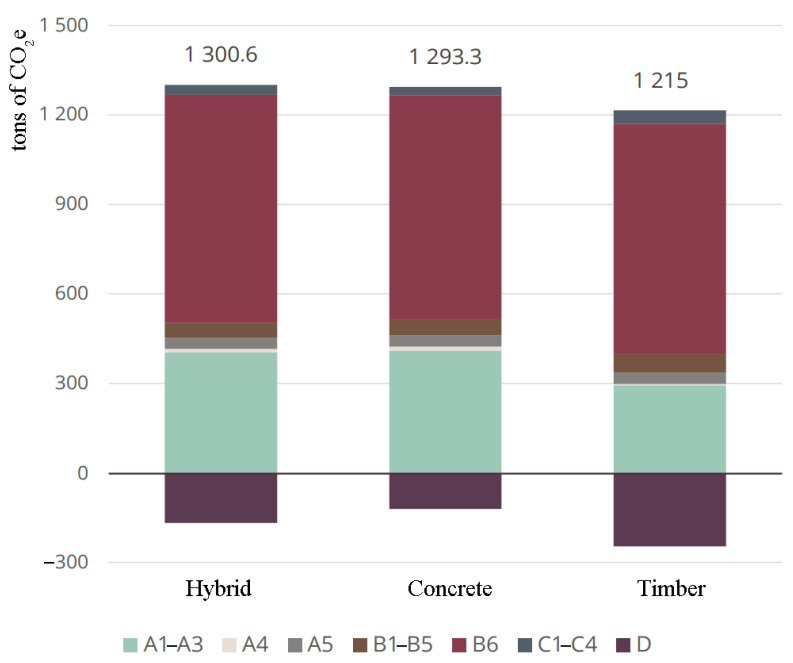
Results of the carbon footprint comparison as a bar chart.

**Figure 11 ijerph-19-00774-f011:**
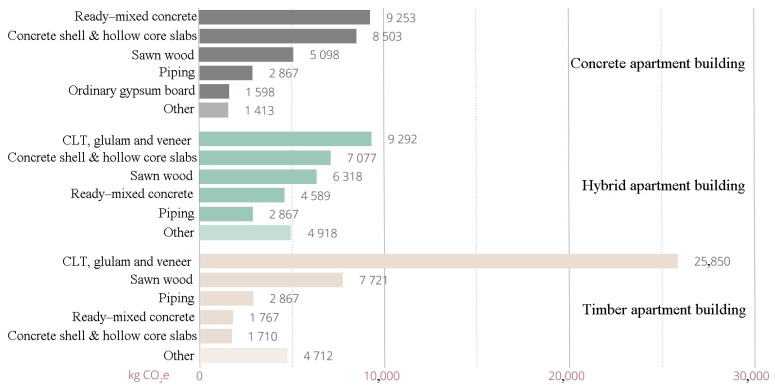
Distribution of carbon footprint by material in end-of-life stages C1–C4.

**Figure 12 ijerph-19-00774-f012:**
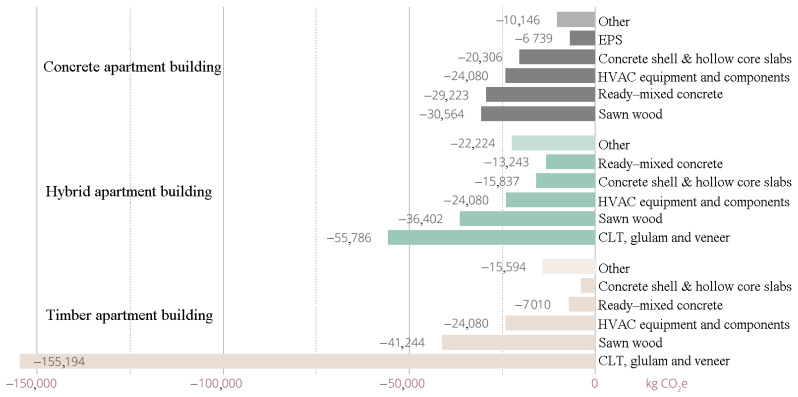
Distribution of carbon footprint by material in module D beyond the building life cycle.

**Figure 13 ijerph-19-00774-f013:**
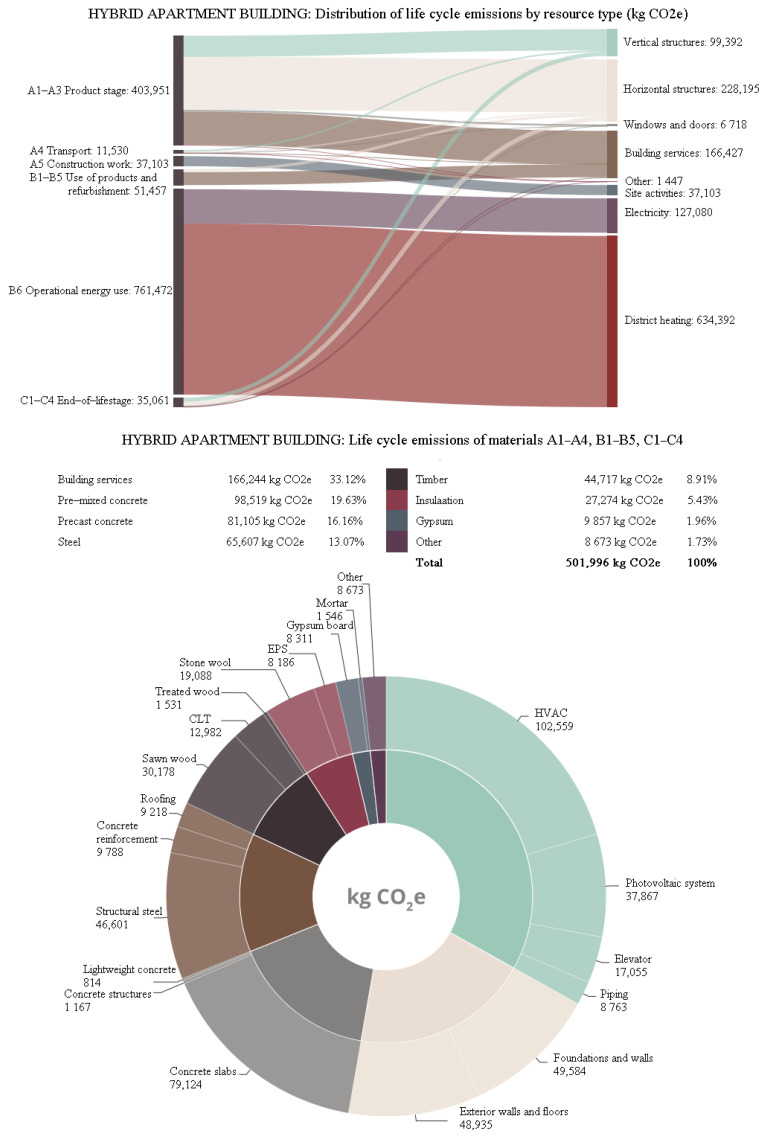
Distribution of the carbon footprint of the hybrid apartment building.

**Figure 14 ijerph-19-00774-f014:**
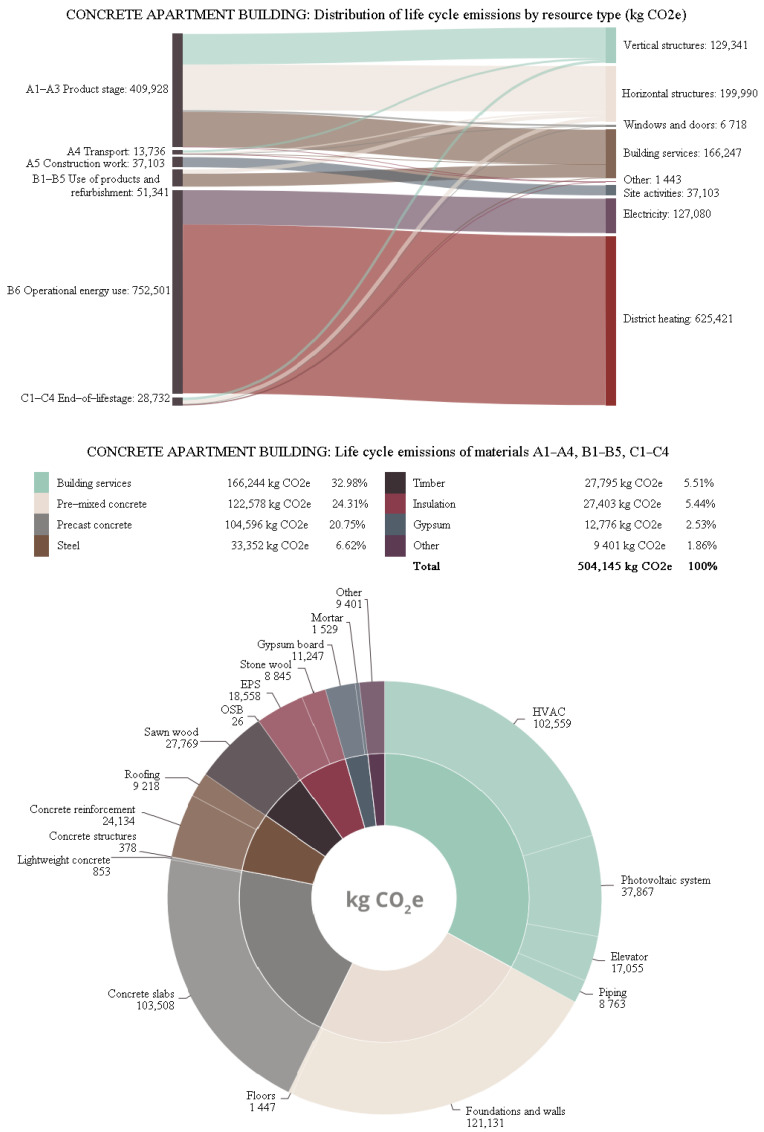
Distribution of the carbon footprint of the concrete apartment building.

**Figure 15 ijerph-19-00774-f015:**
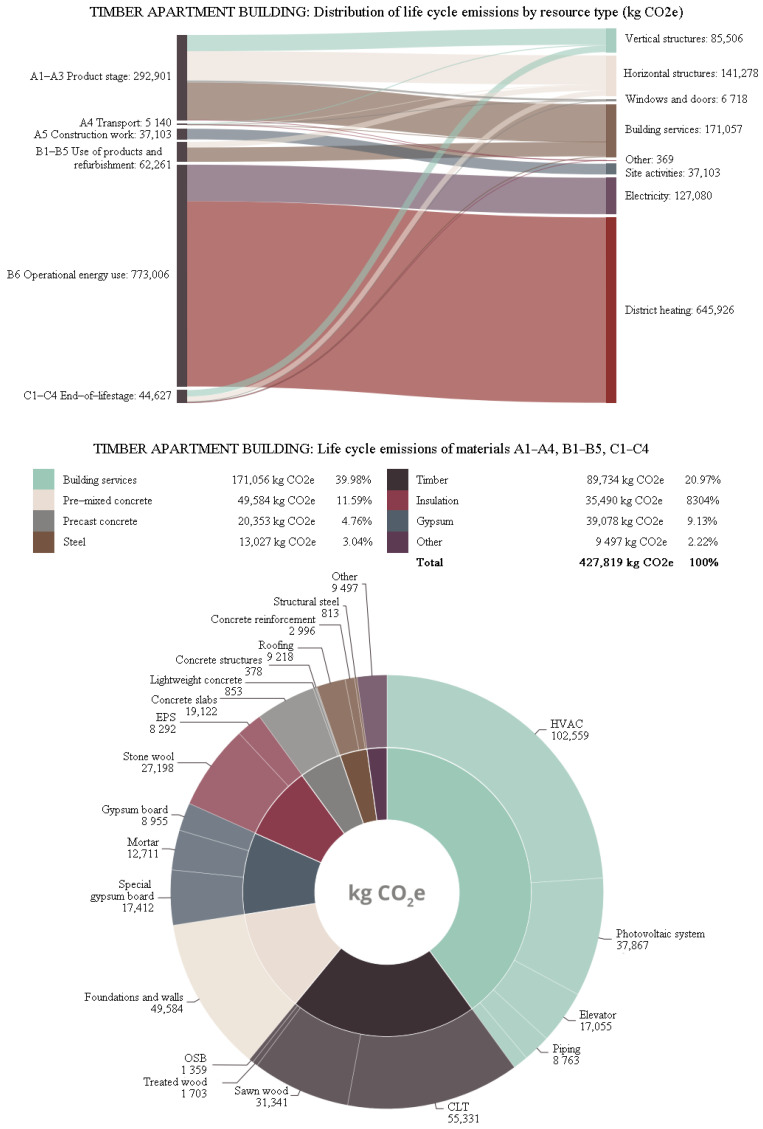
Distribution of the carbon footprint of the timber apartment building.

**Figure 16 ijerph-19-00774-f016:**
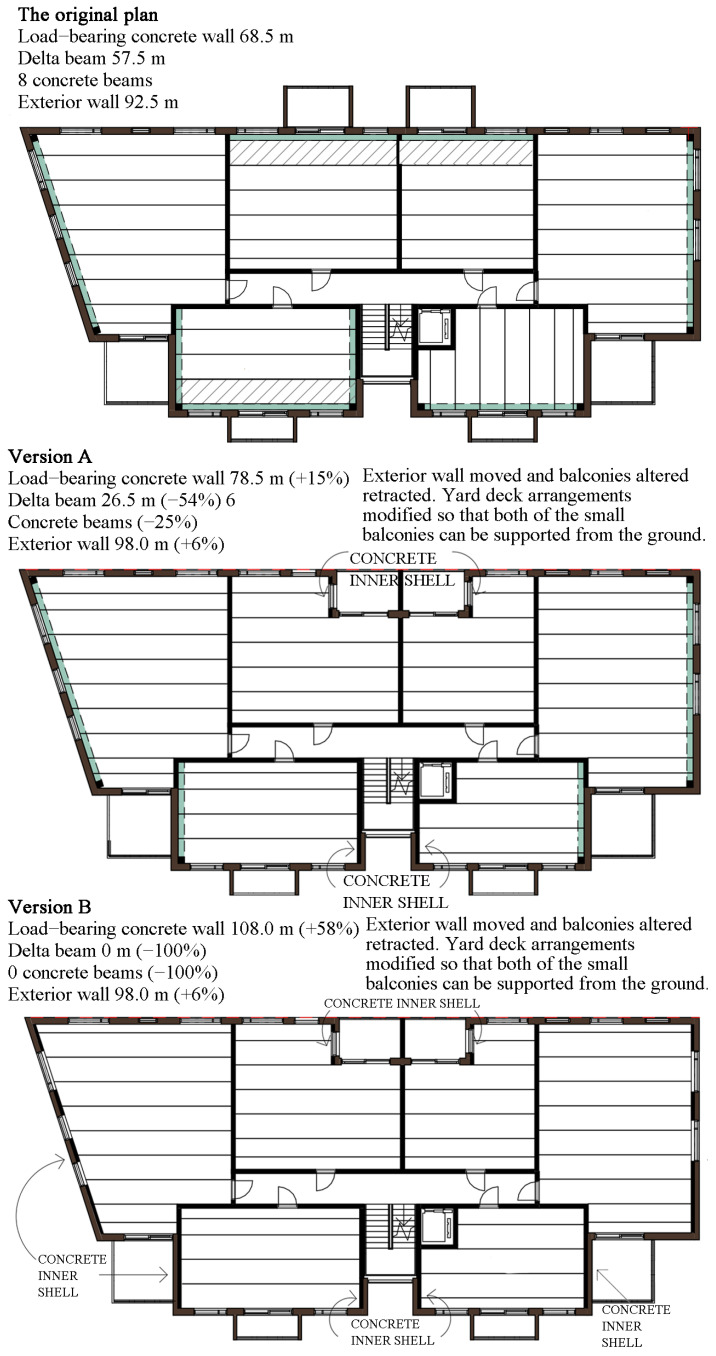
Alternative hybrid apartment building frame solutions to the reduction of carbon footprint.

**Table 1 ijerph-19-00774-t001:** Integration options based on software and formats provided by One Click LCA.

Integration	Notes
Industry foundation classes (IFCs)	ISO 16739/IFC 2 × 3 & IFC4
Autodesk Revit Architectural/Structural Model	v.2016, 2017, 2018 as native plug-ins
IES-VE	v.2017 Feature Pack 4 or higher
Graphisoft ArchiCAD	v.18, 19 as native plug-ins
Tekla Structures	v.2016 as a native plug-in
Simplebim and Naviate Simple BIM 5.0	BIM v.5.0 or higher
DesignBuilder 5.1	v.5.1 or higher
Excel and CSV formats	quantity take-off or costing import
gbXML	Supported, e.g., by IES-VE
Custom integrations from XML, JSON, web services, and other sources	

**Table 2 ijerph-19-00774-t002:** Energy calculation results (with the difference compared with those of the hybrid apartment).

Form of Energy	Hybrid	Concrete	Timber
District heating (kWh/a)	178,200	175,680 (−1.4%)	181,440 (+1.8%)
Electricity (kWh/a)	52,590	52,590	52,590

**Table 3 ijerph-19-00774-t003:** Results of carbon footprint comparison (kg CO_2_ e) (the percentages in parentheses show the difference from those of the hybrid apartment).

Module	Hybrid	Concrete	Timber
A1–A3 Product stage	403,951	409,932 (+1.5%)	292,901 (–27.5%)
A4 Transport	11,529	13,736 (+19.1%)	5140 (–55.4%)
A5 Construction work	37,103	37,103	37,103
B1–B5 Use of products and Refurbishment	51,457	51,341 (–0.2%)	62,261 (+21.0%)
B6 Operational energy use	761,472	752,501 (–1.2%)	773,006 (+1.5%)
C1–C4 End-of-life stage	35,061	28,732 (–18.1%)	44,627 (+27.3%)
In total	1,300,573	1,293,345 (–0.6%)	1,215,038 (−6.6%)
D Beyond the building life cycle	–167,572	–121,058 (–27.8%)	–245,590 (+47.2%)

## Data Availability

Not applicable.

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
