# Peer review of "Comparative Study on Life-Cycle Assessment and Carbon Footprint of Hybrid, Concrete and Timber Apartment Buildings in Finland"

_ijerph, 2022, doi:10.3390/ijerph19020774_

Round 1
Reviewer 1 Report
The research presented in this manuscript is rigorously carried out and clearly described. The technical baseline, and theoretical framework are detailed and up-to-date. The results are very well explained and linked to the conclusions. However, prior to publication of the article I recommend that the authors make three additions:
- Bring out more clearly from the literature review the innovative contribution of this study, possibly highlighting a gap that this research fills.
- Better justify the choice of the One Click LCA tool, possibly citing some previously published studies that have used it. In fact, among the "experts", One Click LCA is certainly considered easy to use, but not as accurate and performing as other commercial software such as GaBi and SimaPro.
- Finally in the conclusions the authors should discuss, in addition to the implications for practitioners and the limits of the research, also the theoretical implications and the innovative contribution given to the literature by the results obtained.
Reviewer 2 Report
The paper tackles an important topic, life cycle assessment of the carbon footprint of buildings in a local study case in Finnland, where the timber resources may pose an option to reinforced concrete. The focus lays however on the method and not on the case study, similar building types being possible also in other countries with timber resources. It might prove useful also in countries with half-timber buildings, which are, among others, earthquake resistant.
The abstract needs improvement. It shall show the context and the methods in addition to the results, and also the conclusions. Therefore the results may be contracted and without abbreviations.
The introduction is generally good and provides numerous relevant references. However, I missed the discussion on the environmental footprint since its definition in Habitat II.
The materials and methods are well described, but the concept of hybrid in the hybrid building could be better detailed. The methods are good, providing also alternatives to reduce the footprint.
The discussion section is therefore relevant and the conclusions are supported by the results. It would be helpful to discuss how the software works with other CAD than autoCAD Revit (ex. archiCAD).
If the above concerns are addressed, I can recommend the paper for publication.
Reviewer 3 Report
Very interesting study which offers a case study approach to LCA method to assess the environmental impact of the construction industry.
The manuscript is well written with a structured content. The results have been clearly presented with the assumptions and limitations of the study.
The only thing I would recommend to the authors is to consider to write the research and practical implications of the study. The statement that starts with "The results are believed ....." does not sufficiently provide the implications.
